# Lightweight Sheep Head Detection and Dynamic Counting Method Based on Neural Network

**DOI:** 10.3390/ani13223459

**Published:** 2023-11-09

**Authors:** Liang Wang, Bo Hu, Yuecheng Hou, Huijuan Wu

**Affiliations:** 1Department of Electronic Engineering, School of Information Science and Engineering, Fudan University, Shanghai 200438, China; wangliang@imu.edu.cn; 2College of Electronic Information Engineering, Inner Mongolia University, Hohhot 010021, China

**Keywords:** dynamic counting, counting sheep, improved SSD, DeepSort

## Abstract

**Simple Summary:**

In the realm of intelligent animal husbandry, the fundamental underpinning of intelligence lies in the capability to identify individual livestock and perform automatic headcounts. Presently, many farming operations still rely on manual counting methodologies, which exhibit notable deficiencies, particularly when confronted with the challenges posed by the substantial populations of sheep and the frequent need for counting. Manual counting is marred by inefficiency and susceptibility to errors such as duplication and omission, thereby presenting formidable hurdles. In response to these challenges within the domain of animal husbandry, this research introduces a deep neural network model that leverages contactless computer vision technology for the automatic detection and enumeration of sheep. We have systematically deployed a series of advanced and efficacious strategies to enhance the model’s performance. Empirical investigations substantiate that our approach can proficiently and accurately automate the counting of sheep within practical farming environments. This method holds significant promise for the intelligent management of sheep farms and possesses adaptability for application across diverse livestock types, underscoring its practical utility. In essence, this study furnishes an effective and practical solution that advances precision and automation in animal husbandry.

**Abstract:**

To achieve rapid and precise target counting, the quality of target detection serves as a pivotal factor. This study introduces the Sheep’s Head-Single Shot MultiBox Detector (SH-SSD) as a solution. Within the network’s backbone, the Triple Attention mechanism is incorporated to enhance the MobileNetV3 backbone, resulting in a significant reduction in network parameters and an improvement in detection speed. The network’s neck is constructed using a combination of the Spatial Pyramid Pooling module and the Triple Attention Bottleneck module. This combination enhances the extraction of semantic information and the preservation of detailed feature map information, with a slight increase in network parameters. The network’s head is established through the Decoupled Head module, optimizing the network’s prediction capabilities. Experimental findings demonstrate that the SH-SSD model attains an impressive average detection accuracy of 96.11%, effectively detecting sheep’s heads within the sample. Notably, SH-SSD exhibits enhancements across various detection metrics, accompanied by a significant reduction in model parameters. Furthermore, when combined with the DeepSort tracking algorithm, it achieves high-precision quantitative statistics. The SH-SSD model, introduced in this paper, showcases commendable performance in sheep’s head detection and offers deployment simplicity, thereby furnishing essential technical support for the advancement of intelligent animal husbandry practices.

## 1. Introduction

In contemporary agricultural practices, sheep farming constitutes a pivotal domain. Nevertheless, the inevitability of sheep mortality or loss during grazing necessitates vigilant monitoring, enabling the timely implementation of preventive measures. Presently, large-scale sheep breeding continues to rely on conventional manual counting methods, characterized by substantial time consumption and an augmented labor burden upon herders. Concurrently, recent years have witnessed significant advancements in artificial intelligence, neural networks, machine learning, and related technological domains. These developments have spurred extensive research and practical applications in target counting via deep learning across diverse domains, encompassing pedestrians [1], vehicles [2], plants [3], and animals [4]. Given the notable advancements in technology, there emerges a considerable scope for research and practical application in the exploration of dynamic vision systems designed for the real-time enumeration of sheep. Such systems hold the promise of addressing longstanding challenges in the domain of sheep farming, with implications extending to enhanced livestock management and agricultural efficiency.

Several researchers have investigated the utility of lightweight neural network models in the context of animal husbandry. In a study conducted by Song et al. [5], an enhanced YOLOv3 model was employed to identify 20 adult Sunit sheep, achieving an impressive mean Average Precision (mAP) of 97.2%. Despite substantial efforts to reduce the model size from its original 235 MB to 61 MB, aimed at mitigating computational costs, the resultant recognition model still possessed a considerable number of parameters, which posed challenges for deployment on mobile devices. Furthermore, the study was constrained by a relatively small dataset of experimental sheep, limiting the ability to comprehensively capture intricate facial features within the sheep facial dataset. Hitelman et al. [6] harnessed the ResNet50V2 model in conjunction with the ArcFace loss function to train facial images of 81 young Assaf sheep, achieving an average recognition accuracy of 97%. Nevertheless, the model size of ResNet50V2 amounted to approximately 98 MB, primarily due to the excessive model parameters, hampering its suitability for deployment on resource-constrained mobile devices. Yang et al. [7], guided by the RetinaFace detection model, implemented parameter reduction via dilated convolution and further enhanced the detection performance via the optimization of the network’s loss function. This culminated in the development of a lightweight sheep detection model. Another study by Billah et al. [8] involved the assembly of a dataset comprising 3278 goat images, incorporating both open-source and manually captured facial images of 10 dairy goats. The YOLOv4 model was leveraged for facial recognition, yielding a commendable recognition accuracy of 96.4%. Nonetheless, the YOLOv4 model exhibited a substantial model size of 244 MB, presenting drawbacks in terms of model size and recognition speed. It is noteworthy that both YOLOv3 and YOLOv4, as earlier iterations within the YOLO series, remained relatively large in model size despite demonstrating commendable performance in sheep facial recognition tasks, rendering them less amenable to practical deployments in sheep facial recognition applications. Currently, there is a conspicuous dearth of research focused on developing lightweight sheep detection and recognition models, as well as the design of mobile systems tailored for such applications. It is apparent that further investigation and refinement in this domain are warranted.

Chae et al. [9] addressed the challenge of detecting cattle mating postures. Building upon the YOLOv3 framework, they introduced specific upper convolution and upper sampling layers, resulting in a network architecture characterized by enhanced detection accuracy, thereby accomplishing the task of cattle mating posture detection. In a related endeavor, MüCher et al. [10] embarked on a multifaceted mission encompassing cattle detection, recognition, and posture classification. They amassed top-view images of cattle and leveraged the Application Program Interface of the Nanonets software tool (5.4) to achieve comprehensive cattle detection, recognition, and posture classification objectives.

Livestock quantity assessment can be categorized into static counting and dynamic counting, contingent on distinct operational conditions. In scenarios where the entirety or a substantial portion of livestock appears within a single image frame, the static counting approach proves effective for quantifying the livestock population. In the realm of static counting methods, Sarwar et al. [11,12] conducted aerial surveillance using Unmanned Aerial Vehicles (UAVs) to acquire images of sheep at varying altitudes. Employing both single-stage and two-stage target detection networks, they adeptly detected diminutive target sheep and subsequently computed their numbers within the images, thereby facilitating the static counting process. Additionally, Tian [13] devised a comprehensive sheep detection and counting system. This system incorporated the design and fabrication of a video capture circuit to capture video imagery. Leveraging the YOLO series target detection algorithm, sheep were systematically detected and quantified within the video frames, yielding valuable static counting results.

In scenarios where capturing the entirety or a predominant proportion of livestock within a single image proves unfeasible, necessitating temporal video analysis for statistical purposes, dynamic counting methods become imperative. The crux of dynamic counting resides in the effective deployment of target detection and tracking technologies. Li et al. [14] conducted comprehensive livestock tracking by installing cameras within livestock movement pathways, capturing video footage of sheep. Sheep head detection was adeptly achieved through the implementation of the YOLOv3 algorithm, and dynamic tracking of sheep was accomplished using the DeepSort tracking algorithm. The quantification of sheep transit through designated counting lines was subsequently facilitated. Kim et al. [15] harnessed the NVIDIA Jetson Nano embedded platform for porcine population statistics. Leveraging the lightweight TinyYOLOv4 model as the target detection algorithm, they introduced an enhanced DeepSort algorithm designed to expedite pig tracking. Ultimately, this method yielded precise pig enumeration based on the counting line. Huang et al. [16] embarked on cattle tail tracking endeavors, introducing an enhanced Single Shot Multibox Detector (SSD) network model to facilitate cattle tail detection. Subsequently, the tail tracking of cattle was effectuated through the utilization of an enhanced Kalman filter and the Hungarian algorithm, building upon the initial detection results. Cowton et al. [17] focused on pig tracking within pig housing environments, employing the Faster R-CNN network for pig detection and localization. Drawing from the detection outputs, they adopted both Sort and DeepSort tracking algorithms to accomplish the multi-target pig tracking task. Concurrently, Zhang et al. [18] addressed the singular pig tracking task, employing a 2D camera situated above pig housing facilities to capture overhead pig imagery. Pig detection and localization were realized using the SSD network, whereas the correlation filter algorithm was adeptly employed to effectuate single pig tracking.

The advancement of artificial intelligence, notably in the realm of computer vision, has ushered in the widespread application of related technologies in the domain of animal husbandry. In endeavors focused on individual livestock identification, prominent target identification networks such as VGG and ResNet [19] are routinely employed. In the context of detecting the spatial location or critical anatomical components of livestock, target detection networks such as Faster R-CNN [20], the YOLO series [21,22,23], SSD, and others find substantial utility.

In the context of livestock enumeration, the application of target tracking algorithms such as Sort, DeepSort [24,25], and correlation filters assumes paramount significance. It is worth noting that initial forays into related computer technologies were primarily oriented towards human-centric applications. However, the realm of animal husbandry necessitates tailored enhancements to these technologies due to distinct contextual settings, utilization requisites, and target entities. In the utilization of machine vision technology for quantitative livestock assessment, the central challenge resides in achieving both precise and expeditious target detection. The overarching objective of this study is to identify a network configuration characterized by high detection accuracy, swift detection speed, and practical deployability. Building upon the foundation of the SSD architecture, this paper introduces the Sheep’s Head-Single Shot MultiBox Detector (SH-SSD). In conjunction with the DeepSort tracking algorithm, the proposed methodology enables the computation of precise sheep quantity statistics.

This paper presents four noteworthy enhancements building upon the foundation of the SSD network architecture [26]. First, in the pursuit of diminishing network parameter volume, while concurrently elevating detection performance, a novel and improved backbone architecture supplants the original VGG backbone. This upgraded backbone, predicated on the MobileNetV3 network, integrates the Triple Attention (TA) mechanism [27] in lieu of the previously employed SE attention mechanism [28]. Second, a feature branch extraction structure is introduced to enhance network feature extraction performance with minimal parameter increment. This structure serves as the foundation for integrating the Spatial Pyramid Pooling (SPP) module and the Triple Attention Bottleneck (TAneck) module within the network’s neck segment. Third, the Decoupled Head (DCHead) structure is introduced to address and optimize spatial displacement issues encountered during the network’s prediction process. This enhancement substantially improves the network’s prediction capabilities. Lastly, the paper adopts the Dynamic Sample Matching method to refine sample matching procedures, ameliorating anomalies in the matching process. These four advancements collectively contribute to the enhanced performance and capabilities of the proposed network model.

## 2. Materials and Methods

### 2.1. Dataset

The data collection process in this study encompasses two distinct components: (1) the network data collection phase involved the acquisition of a dataset comprising 5735 images featuring various sheep. These images were sourced from internet repositories and video frames; (2) the pasture data acquisition segment was conducted in the Etuoke Banner region of Erdos City, Inner Mongolia. The data were gathered at a passage utilized by sheep when entering and exiting a sheepfold. The passage had a width of 2 m, enabling the simultaneous traversal of four sheep. The camera employed for data collection was the DS-NACN54220I-DGI intelligent spherical camera, operating under natural lighting conditions and offering a resolution of 1920 × 1080 pixels. Following the extensive capture of sheep motion video spanning approximately 230 min, a rigorous screening process yielded a corpus of 11,624 valid images extracted from video frames. These images were uniformly saved in JPG format for subsequent analysis. A representative subset of this dataset is visually depicted in Figure 1.

In this study, we employed the LabelImg tool for the meticulous annotation of our dataset. The chosen annotation format adhered to the PASCAL VOC standard, with labeling files adopting the .xml file type. The focus of the annotation encompassed both the frontal and lateral aspects of the sheep’s head, inclusive of its horns. To ensure precision, the annotation frames were meticulously adjusted to closely align with the contours of the sheep’s head, and the designated label was uniformly set as “sheep.” The resultant dataset was comprehensively partitioned into distinct subsets, comprising 10,359 images designated for training, 3500 images for validation, and an additional 3500 images for testing, thereby yielding a grand total of 17,359 images. To comprehensively evaluate the model’s performance in handling small targets and its ability to detect densely occluded targets, two specialized subsets were extracted from the test dataset. The first subset comprised 1156 images specifically designed to assess the model’s performance in detecting small targets. Simultaneously, a second subset, containing 1114 images, was curated to evaluate the model’s detection capabilities in scenarios characterized by significant occlusion.

In pursuit of enhancing the network’s detection capabilities while mitigating the risk of overfitting, several dataset modifications were implemented. These modifications encompassed uniform resizing, color adjustments, random horizontal flipping, and random clipping operations. Specifically, the image dimensions were standardized to 640 × 640 pixels, and the three channels comprising the color fill components were set to values of 114, 114, and 114, respectively. These measures collectively aimed to refine the dataset’s characteristics and enhance its suitability for robust network training and performance evaluation.

### 2.2. Methods

The methodology for sheep enumeration encompasses three integral components: target detection, target tracking, and statistical counting, as elucidated in the illustrative flowchart depicted in Figure 2. In the context of target detection, the SH-SSD network was harnessed as the primary tool. Subsequently, target-tracking procedures were executed using the application of the DeepSort algorithm. The final step involved statistical counting, which was conducted using the Streak method.

In response to the issue of network lightweighting, this study has undertaken substantial enhancements and lightweighting measures to the existing SSD network architecture. These improvements encompass the substitution of the original VGG-based backbone network with an enhanced MobileNetV3 architecture. This architecture, encapsulated through the introduction of the Triple Attention (TA) block, has significantly reduced the overall system parameter count. Additionally, a novel feature branch structure is introduced, serving as the foundation for the construction of the network’s neck, which integrates both the Spatial Pyramid Pooling (SPP) module and the Triple Attention bottleneck (TAneck) module. The network’s head is meticulously engineered, employing the Decoupled Head (DCHead) module. Furthermore, the training regimen entails the adoption of a Dynamic Sample Matching approach, supplanting the conventional fixed matching methodology. Substantial optimizations are also implemented, encompassing adjustments to the size and quantity of feature maps and anchors. In the context of target tracking, the industry-standard DeepSort network is employed, ensuring robust tracking capabilities. For statistical counting, the Streak method is employed, which involves the establishment of predetermined access lines within the video frames. The center point of the sheep’s head frame is designated as the sheep’s positional reference. The counting protocol hinges on evaluating the positional relationship between the sheep’s location and the designated access line. Whenever a sheep is determined to have traversed the access line, an increment of one is applied to the count.

#### 2.2.1. SH-SSD Network

The SH-SSD network, meticulously outlined in this study, comprises three core components: the backbone network, the neck network, and the head network, all elaborated upon in Section 2.2.2, Section 2.2.3 and Section 2.2.4, respectively. The network’s operational workflow is visually represented in Figure 3. The initial segment features an enhanced backbone network rooted in MobileNetV3 enhancements. The subsequent section entails the neck network, a construct comprising SPP and TAneck modules, designed on the foundation of the feature branch structure. Through this, high-level features are derived via profound processing of the underlying features. Finally, the third segment encompasses the head network, which is constituted by a DCHead module responsible for predicting both the network’s location information and classification scores. An NMS module is further employed to sift through redundant detection results, culminating in the extraction of optimal prediction outcomes.

#### 2.2.2. The Lightweight Backbone Network

The original SSD network’s backbone relies upon the VGG network, which, despite its historical significance, exhibits limitations in terms of feature extraction efficacy, network parameter volume, and computational speed, thereby falling short of the prerequisites for the novel network design. In response, this study capitalizes on the pivotal Bottleneck (Bneck) module within the MobileNetV3 network. Herein, we ingeniously amalgamate the Bneck module with the Triple Attention (TA) block, culminating in the inception of the TAneck module. This innovative module seamlessly supplants the traditional Bneck module, consequently yielding an enriched and fortified MobileNetV3 backbone architecture, adept at meeting the discerning demands of our new network design.

Attention mechanisms have gained widespread utilization across various neural network architectures. Notably, the MobileNetV3’s core module, Bneck, previously harnessed the Squeeze-and-Excitation (SE) block for feature enhancement [29,30]. In our pursuit of further elevating the network’s feature extraction capabilities, we opted to replace the SE block with the Triple Attention (TA) block. The TA block has demonstrated superior attentional efficacy and parameter efficiency when compared to its SE counterpart.

The TA block comprises three distinctive branches, as schematically illustrated in Figure 4. The first branch is designated to furnish spatial attention information. This entails the input data undergoing operations that encompass extracting both the maximum and average values along the channel dimension (altering the data’s channel count from C to 2). Subsequently, a 7 × 7 convolution kernel is employed to amalgamate the maximum and average values, followed by batch normalization and activation via the Batch Norm and Sigmoid layers, respectively. Ultimately, the resulting spatial weights are integrated into the input data through element-wise multiplication.

Branch 2 focuses on supplying attention to information pertaining to width variations within distinct channels. Its operational paradigm closely parallels that of Branch 1, with two notable distinctions. First, the extraction of maximum and average values is executed along the height dimension. Second, an intermediate permute layer is introduced before the second convolutional layer and following the sigmoid layer, primarily serving to reorganize the data within the program (H × W × C → C × W × H → H × W × C), thereby streamlining subsequent convolutional operations.

Branch 3, in contrast, is oriented towards delivering attention to information associated with height variations across different channels. Its point of divergence from Branch 2 resides in the extraction of maximum and average values along the width dimension. The ultimate output of these three branches is amalgamated via averaging.

The TAneck module, as depicted in Figure 5a, is a composite structure comprising Depthwise Separable Convolution (DWConv) [31], the TA block, the Convolution-Batch Normalization-ReLU6 (CBR) block, and a residual connection. DWConv represents a convolution operation that operates at the same level as conventional convolution (with the PyTorch parameter “Groups” equating to “In_channels”). DWConv significantly reduces network parameters while accelerating network operations efficiently. The TA block, a pivotal component of TAneck, distinguishes itself by exhibiting both a reduction in parameter count and superior attentional efficacy compared to alternative attention blocks, such as the Convolutional Block Attention Module (CBAM) [32] and the Squeeze-and-Excitation (SE) block.

Within the configuration illustrated in Figure 5b, the CBR module encompasses a convolutional layer, a normalization layer, and the Rectified Linear Unit 6 (ReLU6) activation function. To further enhance the module’s feature extraction prowess, a structural provision for residual connection is incorporated, facilitating the more effective preservation of critical data information. It should be noted that both the residual connection and the TA block are considered optional elements. The inclusion of a residual connection hinges on the congruence between the shapes of the input data (H × W × C) and the output data (H’ × W’ × C”). The criteria for opting for a residual connection and a TA block adhere to the design principles delineated in the MobileNetV3 network settings.

The backbone architecture employed in this study bears a resemblance to the initial six layers of MobileNetV3. However, a notable distinction lies in the utilization of the TAneck module as a foundational component. A schematic representation of the backbone network structure is illustrated in Figure 5c. It is constructed by interleaving the Convolution-Batch Normalization-ReLU6 (CBR) block and the TAneck module. The data undergoes an initial transformation facilitated by the CBR block. Following this, six TAneck modules are employed, yielding the generation of low-level features designated as output1, output2, and output3. These features emanate from the three distinct branches at layers C3 and C6. Subsequently, these low-level features are channeled into the respective branches of the neck network for further processing. In terms of the parameter configuration within the TAneck module, ‘k’ and ‘s’, respectively, denote the convolution kernel size and the stride employed in Depthwise Separable Convolution (DWConv). ‘C’ signifies the number of expansion channels. The ‘AT’ and ‘Res’ flags correspond to the utilization of the TA block and the incorporation of a residual connection. Herein, ‘T’ signifies True, whereas ‘N’ signifies False.

#### 2.2.3. Neck Network

The original SSD network does not incorporate a neck network. In order to enhance the network’s detection accuracy and enrich the prediction feature map with more pertinent information, the inclusion of a neck network becomes imperative. This research introduces a feature branch framework, serving as the foundation for the assembly of the neck network. This neck network is constructed by amalgamating Spatial Pyramid Pooling (SPP) modules with Triple Attention neck (TAneck) modules. Consequently, the neck network plays a pivotal role in enhancing the overall detection accuracy of the network.

To enhance target detection precision and augment the feature map dataset, we incorporate the Spatial Pyramid Pooling (SPP) module into our framework. In essence, expanding the neural network model’s width can be likened to enhancing the granularity of network layers. Consequently, the integration of this module within the network diversifies the information acquired by the network, leading to improved ultimate detection accuracy. The SPP module comprises three distinct maximum pooling branches, as depicted in Figure 6. Input feature data traverses using these three maximum pooling layers, each operating at different scales. The outcomes are subsequently concatenated with the input feature map along the channel dimension. This amalgamation effectively fuses global and local features of varying scales, yielding more enriched and informative feature maps.

In contemporary detection networks, the conventional structure of the neck network predominantly relies on feature fusion mechanisms, exemplified by the commonly employed Feature Pyramid Network (FPN) module [33]. FPN integrates high-level features with their low-level counterparts to address the issue of diminished semantic information in the latter. However, it is worth noting that high-level features themselves tend to contain less detailed information post deep convolution layers, which subsequently limits the inclusion of nuanced semantic information concerning small targets upon their fusion with low-level features. Moreover, this fusion process introduces the information utilized for large target prediction into the feature map intended for small target prediction, thereby potentially causing interference in the prediction of smaller targets. In this study, we introduce a feature branch extraction structure to ameliorate the challenge posed by the scarcity of semantic information in low-level features.

The feature branch structure is visually represented in Figure 7a. Input data traverse via the backbone network, yielding low-level features across three distinct branches. Subsequently, these low-level features undergo extraction via their respective feature branches, culminating in the acquisition of high-level features characterized by varying scales. Ultimately, these high-level features are directed to the head network for prediction purposes. Importantly, the feature branch structure effectively circumvents issues associated with the amalgamation of target information across disparate scales, a problem often encountered in feature fusion structures. Moreover, it accomplishes the objective of enhancing the richness of semantic information pertaining to the targets.

In this study, capitalizing on the feature branch structure as a foundation, the construction of the neck network involves the incorporation of distinct blocks, namely the CBR block, DW-CBR block, SPP module, and TAneck module. Specifically, the DW-CBR block, depicted in Figure 7b, encompasses Depthwise separate convolution operations (DWConv), data normalization via batch normalization (BN), and the utilization of ReLU6 as the network’s activation function. Detailed descriptions of the CBR, TAneck, and SPP modules are provided in Section 2.2.2 and Section 2.2.3, respectively. The architecture of the neck network, as illustrated in Figure 7c, involves the integration of inputs from three branches, originating from the outputs of the backbone network, namely backbone output1, backbone output2, and backbone output3.

Branch1 comprises a sequence of 10 modules, with the initial five modules encompassing two DW-CBR blocks, an SPP module, and two CBR blocks. These DW-CBR blocks facilitate the spatial transformation of features, whereas the SPP module amalgamates local and global spatial features. Subsequently, the CBR modules, acting as post-processing components following SPP, serve to reduce the dimensionality of feature channels, thereby mitigating subsequent computational overhead. The concluding five modules within Branch1 are exclusively composed of TAneck modules, instrumental in orchestrating profound feature transformations to yield high-level feature maps. Branch2 and Branch3 exhibit analogous structures to Branch1, differing primarily in their utilization of four TAneck modules each. Notably, Branch3 omits the SPP component from its configuration.

#### 2.2.4. Head Network

Typically, within the context of target detection networks, a shared feature map is employed for both predicting target locations and target type classifications. Nevertheless, research by Song [34] and Wu et al. [35] has revealed that feature map information relevant to location and classification tasks differs significantly. The use of a unified feature map leads to issues related to spatial misalignment. To address this challenge, this study integrates the DCHead (Decoupled head) module, illustrated in Figure 8a. The DCHead primarily consists of the DW-CBR block and CBR block. The operational sequence initiates with a preliminary transformation of input data via the CBR module. Subsequently, the data are channeled into classification and regression branches. Each branch is constructed from two sets of DW-CBR blocks and CBR blocks, thereby generating feature maps optimized for classification and regression tasks. Ultimately, the predicted values for classification and regression are computed via convolution operations.

The configuration of the head network in this study is visually represented in Figure 8b. Comprising three DCHead modules, these modules correspondingly align with the outputs from the three channels of the neck network. Upon input of the feature map, each DCHead module undertakes the task of adapting the feature map in both the classification and regression channels, thereby refining the feature map for their respective roles in classification and regression. Ultimately, the convolution layer is harnessed to generate predictions for target classification and regression.

## 3. Results

### 3.1. Experimental Environment

The experimental model was developed and assessed within a Windows 10 Professional operating system environment, running on hardware equipped with an i9-10900kf CPU operating at 3.70 GHz. The computational power was bolstered by a 48 GB NVIDIA RTX a6000 graphics card and supported by a substantial 128 GB of RAM. The graphics card operated with driver version 470.63.01 and harnessed CUDA version 11.2 for accelerated computation. Python served as the primary programming language for model development, and PyTorch 1.8.1, a deep learning framework, was instrumental in constructing, training, and testing the neural network. For the Frames Per Second (FPS) assessment, a 48 GB NVIDIA RTX a6000 graphics card was also employed.

The input image dimensions are configured to 640 × 640 pixels. The model’s hyperparameters include a batch size of 32 samples and a total of 300 iterations. The optimization technique employed is Adam optimization, initialized with a learning rate of 0.0005, with scheduled reductions every 100 iterations.

### 3.2. Evaluating Indicator

To assess the algorithm’s efficacy, this study employs the evaluation framework used in the COCO competition. The evaluation metrics employed encompass average detection precision (AP), recall rate (R), frames per second (FPS), and model parameters. Conventionally, AP signifies the average detection precision for a specific class, whereas mean average precision (mAP) signifies the mean average detection precision across all classes. However, in accordance with the COCO evaluation system conventions, this paper refrains from distinguishing between AP and mAP. The calculation of Intersection over Union (IOU) is performed in two modes: at IOU thresholds of 0.5 and 0.5:0.95. The latter mode accounts for AP values computed at 10 IOU intervals ranging from 0.5 to 0.95 with a step size of 0.05. For instance, the AP0.5:0.95 metric reflects the mean AP calculated across the range of IOU thresholds from 0.5 to 0.95 (i.e., 0.5, 0.55, 0.60, 0.65, 0.70, 0.75, 0.80, 0.85, 0.90, and 0.95).

The term “Prediction (P)” serves as a metric for quantifying the precision of the ultimate prediction outcomes pertaining to a specific target class. In essence, it denotes the ratio of correctly identified sheep’s head targets using the model to the overall detected targets, as expressed in Equation (1).
(1)$$P=\frac{TP}{TP+FP}\times 100\%$$
where TP denotes the count of accurately detected sheep’s head instances, whereas FP signifies the tally of targets erroneously identified as sheep’s head.

The recall rate, denoted as “R”, serves as a metric for assessing the extent to which the target detection model has inclusively identified all intended targets. It quantifies the ratio of correctly detected sheep’s head targets to the entirety of the intended targets, as depicted in Equation (2).
(2)$$R=\frac{TP}{TP+FN}\times 100\%$$
where FN indicates the number of missed sheep’s head targets.

The average detection precision (AP) can be understood as the area under the precision-recall (PR) curve, which is expressed mathematically in Equation (3). Furthermore, the small target average precision (AP) pertains to the mean accuracy computed for all targets characterized by an area smaller than 32 × 32 units.
(3)$$AP=\int_{0}^{1}P\left(R\right)dR$$

The metric known as frames per second (FPS) represents the quantity of images that the model can effectively detect within a one-second interval. Notably, a higher FPS value signifies an enhanced real-time performance of the network, which is expressed mathematically in Equation (4).
(4)$$FPS=\frac{M}{S}$$
where M represents the number of images detectable, and S represents the time taken to detect M images (in seconds).

### 3.3. Detection and Analysis of SH-SSD Network

As depicted in Table 1, the original VGG16 backbone in the SSD network has been substituted with MobileNetV3. This substitution has yielded a noteworthy enhancement in detection accuracy, achieving an increase of 0.78%. Moreover, the detection speed has more than doubled, while the network’s parameter count has been reduced by a factor of nearly 8. To further augment the network’s detection accuracy, this study employs the TA block to reinforce the MobileNetV3 backbone. This enhancement results in a substantial 1.51% increase in detection accuracy, with negligible impact on detection speed and network parameters.

As presented in Table 2, this comparative analysis examines the impact of enhancements introduced to the SSD network, including the integration of an additional feature branch, the DCHead module, and dynamic matching. According to the findings in Table 2, it is evident that the incorporation of the DCHead module yields a notable enhancement in average detection accuracy by 1.04%, the SH-SSD network showcases a remarkable capacity for retaining small target information, thus addressing this issue. On the other hand, the adoption of dynamic sample matching optimizes the network training process without modifying the network structure, thus maintaining detection speed and parameters. Following the implementation of dynamic matching, there is a substantial enhancement in average detection accuracy by 1.28%. Furthermore, during the verification testing phase, the feature branch structure is introduced to the lower feature layer. Consequently, this results in an improvement of 0.76% in average detection accuracy.

### 3.4. Comparison with the SOTA Network

In this study, a meticulously curated dataset is employed for the training and evaluation of multiple target detection networks, including YOLOv3-SPP, YOLOv5, Dual Weighting [36], SSD, and the proposed SH-SSD network. The primary objective is to assess their proficiency in detecting small-sized targets and targets exhibiting high levels of density and occlusion. The comparative performance of these target detection networks under varying conditions is visually presented in Table 3. The resultant detection outcomes offer valuable insights into the efficacy of the SH-SSD network in contrast to the other four network variants. Specifically, in scenarios involving diminutive, densely packed, and partially obscured targets, the SH-SSD network consistently excels in delivering precise and high-fidelity recognition.

It is noteworthy that the YOLOv3-SPP detection network exhibits certain limitations, particularly in terms of missed detections. Moreover, both the YOLOv5 detection network and the Dual Weighting network exhibit relatively lower recognition accuracy compared to the SH-SSD network. In contrast to the SSD network operating at an IOU threshold of 0.5, the SH-SSD network demonstrates a noteworthy enhancement in average detection accuracy, realizing a substantial improvement of 6.8%. Moreover, when evaluated across a range of IOU thresholds from 0.5 to 0.95 with increments of 0.05, the SH-SSD network exhibits a remarkable advancement, achieving an impressive 12.15% augmentation in average detection accuracy alongside a commendable 11.21% boost in recall rate. Notably, these substantial performance gains are accompanied by a reduction of 17.47 million network parameters. Overall, this augments the efficacy of the network while maintaining a consistent detection speed. Consequently, when juxtaposed with the performance of the other four network architectures, it becomes evident that the SH-SSD network not only demonstrates superior detection capabilities but also exhibits heightened recognition proficiency in the context of sheep’s head detection, rendering it better suited for real-world applications. A comprehensive summary of the comparative results of the five target detection networks is presented in Table 3.

### 3.5. Dynamic Counting Test of Sheep

To assess the dynamic counting efficacy for sheep, the SH-SSD network was employed for target detection, complemented by the widely adopted DeepSort algorithm for precise target tracking. The counting criteria were determined using a specific marking methodology, which entailed incrementing the statistical result upon the passage of the sheep’s head center point across a predefined red line, oriented from the top to the bottom of the video frame. Figure 9 visually illustrates the counting performance achieved by the SH-SSD network. We systematically augmented the sheep population and conducted a total of 530 randomized experiments, as shown in the scatter plot of Figure 10, achieving precise enumeration in 518 instances and attaining an accuracy rate of 97.74%. The minor counting inaccuracies observed were primarily attributed to the challenges posed by severe occlusion resulting from the high density of sheep in certain scenarios. The findings notably affirm that the utilization of the SH-SSD network as the primary detection mechanism yields accurate sheep counting results.

## 4. Discussion

This study seeks to investigate the detection attributes pertaining to sheep’s head. To this end, the SH-SSD network is developed on the foundation of the SSD network, accompanied by the establishment of a dataset specific to sheep’s head. The efficacy of the SH-SSD network is scrutinized within the context of this dataset. Furthermore, the paper examines the SH-SSD network’s performance in tandem with the DeepSort tracking algorithm for target tracking. The ensuing methodology and outcomes of this research endeavor are succinctly encapsulated as follows.

The incorporation of the Triple Attention mechanism, as a replacement for the SE attention mechanism within the MobileNetV3 backbone network, notably enhances the model’s feature extraction capabilities. Additionally, we introduce a feature branching structure, implemented following observations of inadequate preservation of detailed feature information and the need for semantic enrichment, particularly evident in low-level and middle-level features. Furthermore, the DCHead structure is integrated, serving as the detection head within the network. This addition effectively addresses spatial dislocation issues in both positioning and classification predictions, consequently enhancing the overall detection performance. Lastly, our model training strategy embraces Dynamic Sample Matching, a dynamic optimization approach that adapts subsequent learning labels based on the network’s detection outcomes, thereby fostering superior learning outcomes.

The DeepSort tracking algorithm was employed to conduct a comparative analysis between the original SSD network and the SH-SSD network regarding their performance in sheep quantity statistics. The findings unequivocally demonstrate that the SH-SSD network yields more precise statistical results and enhances the tracking performance of the model. These results underscore the superior suitability of the SH-SSD network within the context of sheep quantity statistical modeling.

Given the expansive scope of investigation concerning the dynamic counting of sheep, several avenues for further enhancement exist in this research domain. The dataset’s scope, encompassing the quantity of images, the diversity of sheep species, and the richness of backgrounds, currently remains constrained. It is imperative that the dataset be continuously augmented in subsequent research efforts. Currently, a bottleneck in speed is discernible within the target tracking module. In future endeavors, we will persist in delving into the target tracking module to ameliorate the ultimate statistical speed.

## 5. Conclusions

In comparison to the conventional SSD network, the SH-SSD network exhibited notably enhanced detection accuracy, achieving 96.11% and 63.41% in the AP0.5 and AP0.5:0.95 metrics, respectively. These figures denote substantial improvements of 6.8% and 12.05%. Notably, the most substantial enhancement was observed in the AP0.5:0.95 metric, signifying a remarkable augmentation in accuracy. These outcomes underscore the superior positioning and classification capabilities of the SH-SSD network model, thereby attesting to its heightened detection performance.

In comparison to other detection networks, which exhibit superior detection accuracy, the SH-SSD network consistently demonstrates commendable detection performance. While its detection speed is marginally slower than that of YOLOv5, it still satisfies real-time detection demands. These findings underscore the suitability of the SH-SSD network for tasks involving sheep’s head detection.

This study centers its investigation on sheep as the primary research subject, with a specific emphasis on addressing issues encompassing dynamic counting and tracking within the detection model. Additionally, it delves into topics such as dataset construction, optimization of convolutional neural networks, and enhancements in tracking and detection accuracy. Our experimental outcomes highlight the practical feasibility of applying tracking results to a single-line counting approach, yielding effective sheep counting. Consequently, we justified confidence in asserting that the automatic sheep-counting methodology outlined in this paper aligns with the requisites of real-world application. This method holds inherent value within the context of the sheep farming industry, while concurrently presenting novel insights for the broader realm of intelligent animal husbandry production.

## Figures and Tables

**Figure 1 animals-13-03459-f001:**
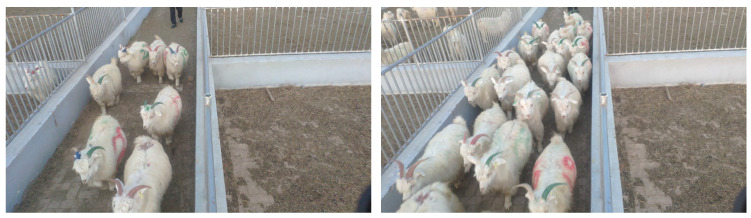
Examples of sheep images.

**Figure 2 animals-13-03459-f002:**
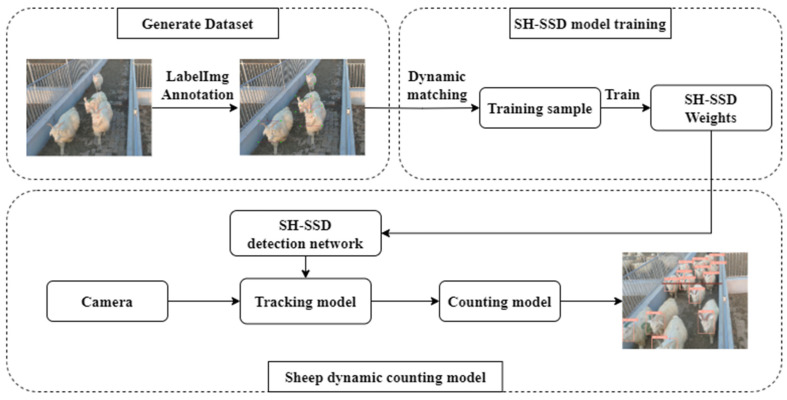
Sheep count flow chart.

**Figure 3 animals-13-03459-f003:**
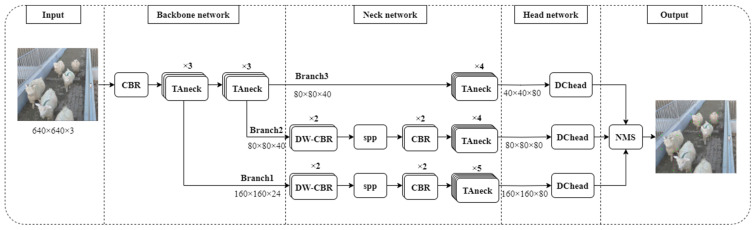
Workflow diagram of the SH-SSD network.

**Figure 4 animals-13-03459-f004:**
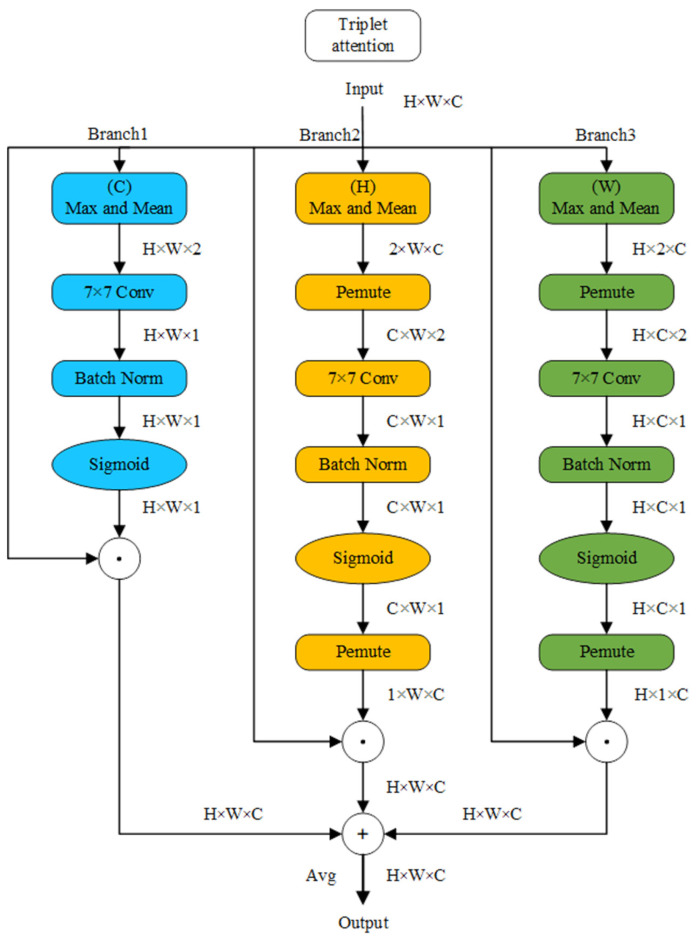
Structure diagram of the Triple attention block.

**Figure 5 animals-13-03459-f005:**
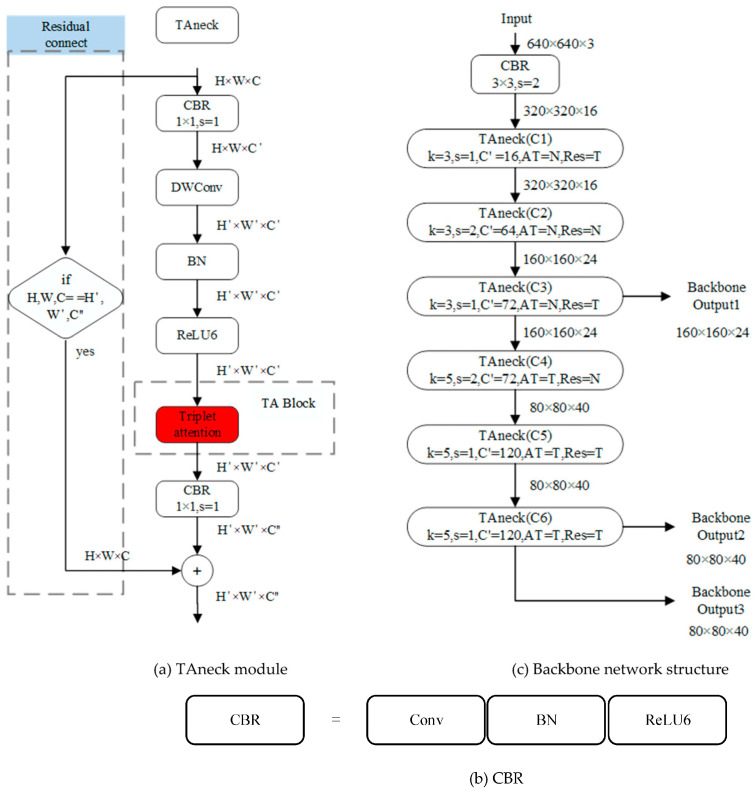
Backbone of SH-SSD structure diagram.

**Figure 6 animals-13-03459-f006:**
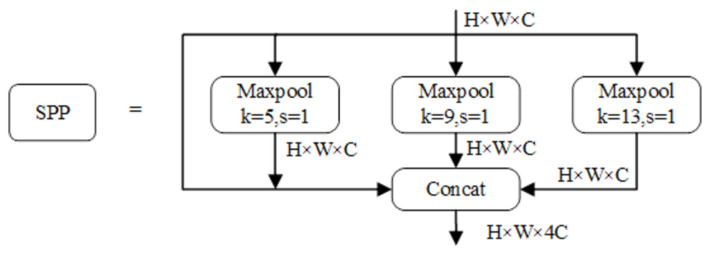
Structure diagram of the SPP module. Maxpool indicates the maximum pool. k indicates the kernel size. s indicates the stride.

**Figure 7 animals-13-03459-f007:**
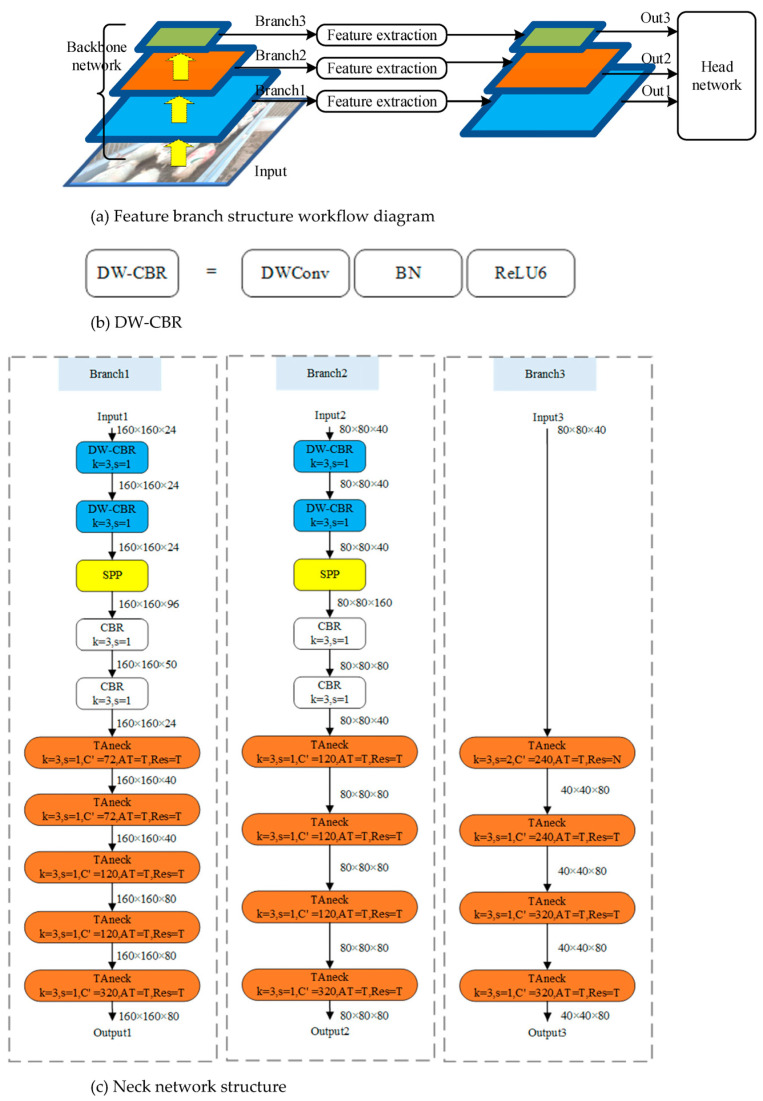
Neck of SH-SSD structure diagram.

**Figure 8 animals-13-03459-f008:**
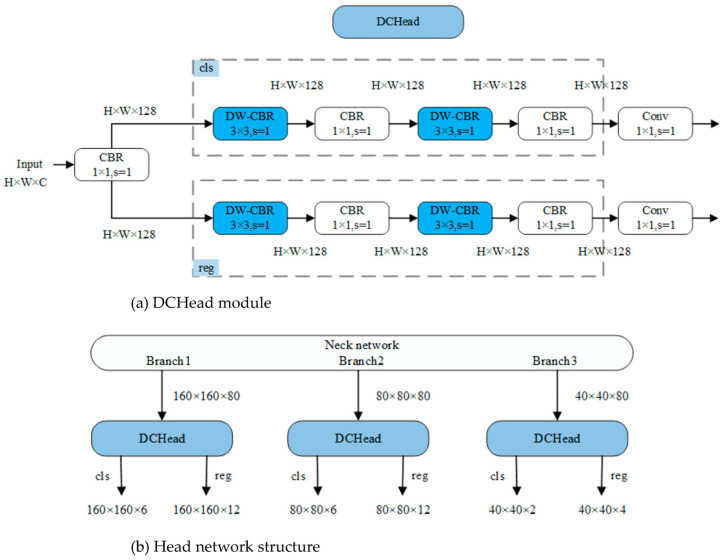
Head of SH-SSD structure diagram.

**Figure 9 animals-13-03459-f009:**
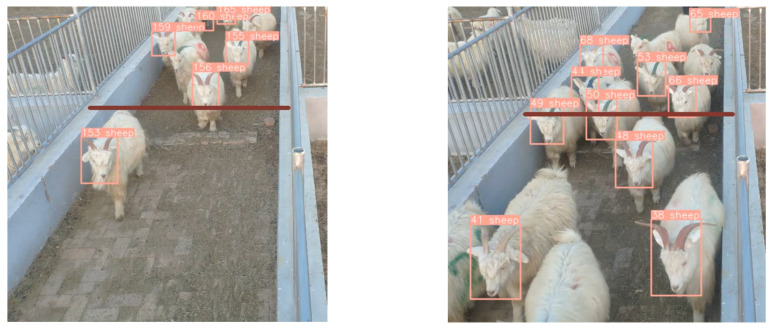
SH-SSD network for quantification.

**Figure 10 animals-13-03459-f010:**
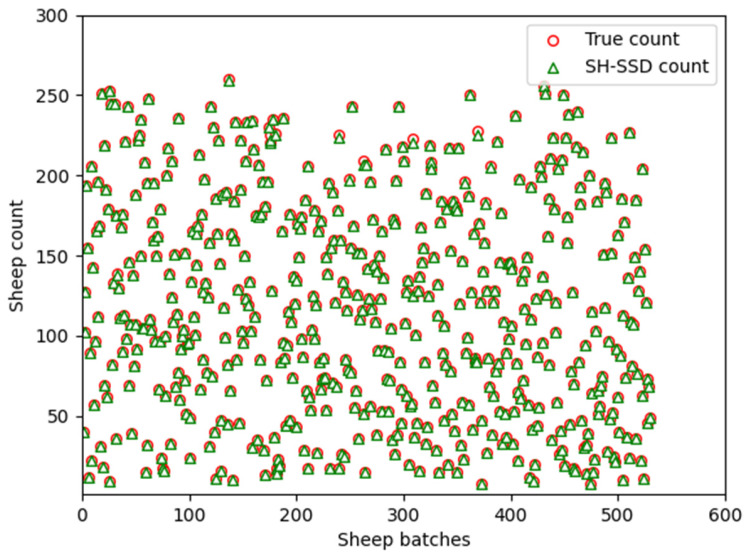
Test results of SH-SSD network.

**Table 1 animals-13-03459-t001:** Comparison effect of different backbone networks.

Backbone	AP0.5/%	FPS	Param/M
VGG16	89.31	91	23.81
MobileNetV3	90.09	193	2.98
TA enhanced MobileNetV3	90.82	190	2.97

**Table 2 animals-13-03459-t002:** An ablation study concerning the enhancements introduced in our research.

Methods	AP 0.5/%
SSD	89.31
SSD + DCHead	90.35
SSD + Dynamic Matching	90.59
SSD + Feature Branch	90.07
SSD + DCHead + Dynamic Matching	92.18
SSD + DCHead + Dynamic Matching + Feature Branch	96.11

**Table 3 animals-13-03459-t003:** Comparison of the effects of SOTA networks.

Network Structure	AP 0.5/%	AP 0.5:0.95/%	APs 0.5:0.95/%	R 0.5:0.95/%	FPS	Param /M
YOLOv3-SPP	91.25	53.63	16.42	62.09	71	62.57
YOLOv5	94.81	59.91	21.01	66.31	90	21
Dual Weighting	95.87	62.93	25.20	70.72	28	48.01
SSD	89.31	51.36	59.86	59.86	91	23.81
SH-SSD	96.11	63.41	27.81	71.07	84	6.44

## Data Availability

The data presented in this study are available upon request from the corresponding author.

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
