# Peer review of "Lightweight Sheep Head Detection and Dynamic Counting Method Based on Neural Network"

_animals, 2023, doi:10.3390/ani13223459_

Round 1

Reviewer 1 Report

Comments and Suggestions for Authors

The paper describes a sheep counting system based on identifying the animals' heads from video images. The paper is well-founded, well-structured and complete. English has no detected faults and almost no typos.

In technical terms, the methodology, tests and results obtained seem solid and obtained in a valid and coherent way.

I am in favor to paper publication, but I think you should address this 5 remarks:

- Your approach  is not the most suitable for animal detection since RFID would allow to really track animals using a set of RFID readers, and to be easily integrated  with other processes like milking robots or eating dispensers.

- The method comparison does not specify detection cost, we just know the number of detectable images per second. Tt allows to compare methods merit, but does not allow to assess system scalability. How many sheep does the system detect per minute? How many sheep does the system detect per frame?

- Paper introduction and conclusions refer repeatedly animal tracking, but your proposal just implements animal detection. Animal tracking would require at least a second instance of your system.

- line 527: “represents the number of images” -> “represents the number of images detectable”

- Typos:

References include a set of typos: are not complete, include some capitalized caracteres enclosed on square brackets .

Consider to remove the 4th order titles (e.g. “1. Spatial pyramid pooling module”). This is not mandatory from the journal template, but I would do it.

Reviewer 2 Report

Comments and Suggestions for Authors

The authors claimed that the SH-SSD model showcases commendable performance in sheep's head detection and offers deployment simplicity.

Their methodologies and findings support the authors' claim. 

I did not find any significant flaw in their study design. However, I found the following minor issues:

- Cited references from 5 to 19 on P2 and P3 are not consistent with the citation format of the journal.

- Figures 1 and 2 on P5 need to have better resolution. 

This manuscript should be accepted after minor issues are addressed.

Reviewer 3 Report

Comments and Suggestions for Authors

The paper introduces a deep neural network model that leverages contactless computer vision technology for the automatic 16 detection and enumeration of sheep. Te method holds significant promise for the intelligent management of sheep farms and possesses adaptability for application across diverse livestock types, underscoring its practical utility.

The work is well structured, the subject is very promising.

Several neural network structures were presented and a comparison was made between them. In comparison to the conventional SSD network, the SH-SSD network exhibited notably enhanced detection accuracy, denote substantial improvements of 6.8% and 12.05%. 

Author Response

    We greatly appreciate the reviewer's positive feedback and acknowledgment of the potential practical utility of our work. We are pleased to hear that you found the structure of the paper to be well-organized and the subject matter promising.

    Your positive comments are highly motivating and reassure us that our research is headed in the right direction. We are committed to further refining and improving our work to meet the high standards of the journal and the expectations of the scientific community.

    Thank you for your encouragement and feedback. We look forward to incorporating your suggestions and continuing to contribute to the field of intelligent animal husbandry through our research. If you have any specific recommendations or comments on how we can enhance the paper further, please feel free to share them.

Reviewer 4 Report

Comments and Suggestions for Authors

The development of monitoring techniques for real-time flock numbers has positively supported sheep farming. The authors tried to improve the tracking algorithm of Deepsort and to lightweight the model. This is of practical interest. The following questions are for the authors.

(1) Since the title of the article emphasizes the lightweighting of the model, the manuscript has to focus on the relevant research progress and the problems of the current technology in the introductory section. At the same time, the methods section should focus on the lightweighting of the model and describe the relevant solutions.

(2) From the acquisition of model training samples, it seems that the training samples come from a single video surveillance scene, and the surveillance time is only 230 minutes. With such a single surveillance scene and such a short sampling time, how to guarantee the generalization of the model?

3) In terms of the logical structure of the paper, the general framework of the model (Fig 8) should be introduced first rather than the modules of the model first, which is extremely unfriendly to readers.

(4) There are serious inconsistent conceptual errors in Figure 2. The essence of the article is to improve the target detection module of Deepsort and retain the trajectory tracking module of Deepsort. Therefore, Deepsort should not appear in Figure 2, otherwise SH-SSD has no meaning of existence.

(5) Deepsort is a very useful and influential model for target detection and tracking. There have been a number of studies to improve the model performance by replacing the target detection module in this model. Why use a relatively old SSD for improvement instead of using a model that is already lightweight and has relatively high initial performance, such as the yolo family?

(6) Please explain the meaning of the specific features represented by each of the 3 Branch1. If you use softMax instead of Sigmoid activation function, can you strengthen the correlation between different features and highlight the most influential part of the attention while weakening the less influential region in your design of the attention function? Has this design been tried and how effective is it?lines 233

(7) The reference format does not match Animals.

(8) The ablation test has to be increased.

(9) The results of the trials in Table 4 are too simplistic and unconvincing. The authors have to perform at least 500 or more randomized detection trials to assess the statistical performance of the model with as many test results as possible.

Round 2

Reviewer 4 Report

Comments and Suggestions for Authors

The authors have provided adequate explanations and revisions based on the reviewing comments. However, the manuscript still needs some minor revisions: (1) the resolution of Figure 3 is insufficient; (2) it would be desirable to standardise the font sizes in all figures; (3) as required by the journal, the main acronyms used in the manuscript should be listed after the conclusion sectionï¼›and (4) the results of Table 4 should preferably be presented in the form of a scatterplot of the observed Vs. predicted values.
